**Data Availability Statement:** All relevant data are within the paper and its Supporting Information files.

# The investigation of antibacterial properties of peptides and protein hydrolysates derived from serum of Asian water monitor (*Varanus salvator*)

**Jitkamol Thanasak**[1]*, **Sittiruk Roytrakul**[2], **Waraphan Toniti**[3], **Janthima Jaresitthikunchai**[2], **Narumon Phaonakrop**[2], **Siriwan Thaisakun**[2], **Sawanya Charoenlappanit**[2], **Rudee Surarit**[4], **Wanna Sirimanapong**[1]

**1** Faculty of Veterinary Science, Department of Clinical Sciences and Public Health, Mahidol University, Nakhon Pathom, Thailand, **2** Functional Proteomics Technology Laboratory, National Center for Genetic Engineering and Biotechnology (BIOTEC), National Science and Technology Development Agency, Pathum Thani, Thailand, **3** Faculty of Veterinary Science, Department of Pre-clinic and Applied Animal Science, Mahidol University, Nakhon Pathom, Thailand, **4** Faculty of Dentistry, Department of Oral Biology, Mahidol University, Bangkok, Thailand

* jitkamol.tha@mahidol.ac.th

## Abstract

It is well known that the Asian water monitors or *Varanus salvator* are both scavengers and predators. They can live and survive in the place that exposed to harmful microorganisms. Most people believe that they have some protected mechanisms to confront those infections. The aim of this study is to determine the antibacterial activities of crude peptides and protein hydrolysates extracted from serum of the *Varanus salvator*. Ten types of bacteria were cultured with crude peptides and protein hydrolysates which were isolated from 21 *Varanus salvator's* serum. The crude peptides showed some interested inhibition percentages against *Enterobacter aerogenes* ATCC13048 = 25.6%, *Acinetobacter baumannii* ATCC19606 = 33.4%, *Burkholderia cepacia* ATCC25416 = 35.3% and *Pseudomonas aeruginosa* ATCC27853 = 25.8%, whereas the protein hydrolysates had some inhibition potential on *Burkholderia cepacia* ATCC25416 = 24.3%. For the rest results of other tests were below 20% of inhibition. In addition, the evidences show that crude peptides have better antibacterial performances significantly than protein hydrolysates on most tested bacteria. Furthermore, antimicrobial peptides prediction shows about 10 percent hit (41/432 sequences). The interpretation shows that the best hit sequence is highly hydrophobic. It may destroy outer membrane of Gram-negative hence prevents the invasion of those bacteria. Altogether, bioinformatics and experiments show similar trends of antimicrobial peptide efficacy from *Varanus salvator*. Further studies need to be conducted on peptide purification and antimicrobial peptide candidate should be identified.

**Funding:** Mahidol University (Award number: NDFR 09/2563 | Recipient: Jitkamol Thanasak). The funders had no role in study design, data collection and analysis, decision to publish, or preparation of the manuscript.

**Competing interests:** The authors have declared that no competing interests exist.

## Introduction

The peptides are thought to be a new group of biomolecules that are effective in inhibiting the growth of resistant (vancomycin-resistant enterococci) and non-resistant pathogenic bacteria (*P. aeruginosa*, *S. typhimurium*, *E. coli*, *S. aureus*, *S. epidermidis*, *L. monocytogenes*). In nature, antimicrobial peptides in plants, animals and humans serve to prevent the invasion of other organisms that cause disease [1]. Antimicrobial peptides that have the ability to kill bacteria, viruses, parasites and fungi, such as "cecropin" from insects, "margainins" from amphibians and "defensins" from mammals, which more than 2,500 different antimicrobial peptides have been discovered from various living organisms [2–8]. Antimicrobial peptides have been reported to have a cationic, anion and hydrophobic residues structure, allowing them to bind to the bacterial cell wall and destroy the bacterial cell wall, leading to the breakdown of the bacteria. In addition, it can also stimulate the immune system to work with the antimicrobial peptide effectively [9,10].

A peptide extracted from bovines, BMAP-28, can resist the growth of both Gram-positive and Gram-negative bacteria such as *Escherichia coli* ATCC25922, *Salmonella typhimurium* ATCC14028, *Pseudomonas aeruginosa* ATCC27853, *Serratia marcescens* ATCC8100, *Staphylococcus aureus* ATCC25923, *S. aureus* Cowan 1, Methicillin resistant *S. aureus* (MRSA), *Staphylococcus epidermidis* ATCC 12228, *Bacillus megaterium* Bm11 and Carbapenam resistant Enterobacteriaceae [11,12]. A peptide extracted from scorpion venom, IsCT, can inhibit the growth of bacteria such as *E. coli* ATCC1682, *P. aeruginosa* ATCC1637, *S. typhimurium* ATCC1926, *S. epidermidis* ATCC1917, *S. aureus* ATCC1621, MRSA CCARM 3543 and MRSA CCARM 3001 [13]. Moreover, BmKn-2, an animal peptide especially extracted from scorpion venom, has shown the inhibitory effect on bacterial growth [11,14,15]. A novel peptide toxin from the Spider *Pardosa astrigera*, Lycotoxin-Pa4a, inhibited both Gram-negative bacteria (*Escherichia coli* KCCM 11234 and *Pseudomonas aeruginosa* ATCC 9027) and Gram-positive bacteria (*Bacillus cereus* KCCM 21366 and *Staphylococcus aureus* KCCM 11335), as well as, some anti-inflammatory properties have been reported [16,17]. In addition, a unicellular organism such as *Saccharomyces cerevisiae* (Baker's yeast) has been considered for antimicrobial properties on bioactive peptides. The results showed that the peptide produced by *Saccharomyces cerevisiae* ATCC 36858 had exerted inhibition activity against both Gram-negative and Gram-positive bacteria, including *Bacillus subtilis* ATCC 23857, *Escherichia coli* ATCC 25922, *Klebsiella aerogenes* ATCC 13048, and *Staphylococcus aureus* ATCC 25923. The isolated antibacterial peptide from *Saccharomyces cerevisiae* might demonstrate its potential as a bio-preservative compound against Gram-negative and Gram-positive bacteria in food manufacturing [18]. The treatment of bacterial cultures with serum from Komodo dragons, *Varanus komodoensis*, showed evidence of reduction in bacterial growth [19]. The cationic antimicrobial peptides, CAMPs, extracted from Komodo dragon plasma had been investigated which 8 of 48 potential CAMPs had ability to against *Pseudomonas aeruginosa* ATCC 9027 and *Staphylococcus aureus* ATCC 25923 [20].

The Asian water monitor, *Varanus salvator*, is a reptile belonging to the family Varanidae [21], which is one of the most common monitor lizards throughout Southeast Asia ranging from India, Bangladesh, Sri Lanka, Indochina and Indonesia [22]. Even though, it is under the Wild Animal Reservation and Protection Act, B.E. 2535 (1992) according to the list of Thailand protected species [23], the Varanus population that has grown so much has generated considerable debate over the management of the use of benefit from this species. By nature, these Varanids are both scavengers and predators. They are often exposed to harmful microorganisms and noxious agents such as heavy metals. Since, they can survive even in dirty places and rarely got the affected by septic infections, it indicates that they have developed some

mechanisms to protect themselves from potential sepsis. Microbiome studies have shown evidences that captive Komodo dragons, *Varanus komodoensis*, share the microbiomes of their skin, saliva, and feces extensively with their environment such as soil, sediment, and water independently [24]. The dietary relationship of oral and gut microbiota in the water monitor lizard, *Varanus salvator*, has also been demonstrated that Proteobacteria and Bacteroidota were the prominent phyla in the oral and gut microbes, which microbial community obtained from aquatic products could help water monitor lizards better adapt to the environment. In addition, the microbiota diversity in the gut was higher than in the oral and the abundance of oral microbiota was higher in captive lizards than in wild lizards [25]. Several bacterial species, both Gram-negative and Gram-positive, have been isolated from the saliva of the Komodo dragons, which were greater in the mouths of wild dragons than captive dragons [26,27]. Some species of bacteria isolated from wild dragon saliva showed anti-bacterial resistance properties that may cause harmful to human and their prey [28]. Furthermore, some bacterial species, including *P. mirabilis*, *A. hydrophila*, *C. freundii*, *E. coli*, *Staphylococcus sp.* and *S. aureus*, isolated from the gut of water monitor lizard, had potential to produced molecules with broad-spectrum antibacterial activities [29]. Moreover, the gut microbiota of *Varanus salvator* might be considered as a potential property of antitumor molecules [30]. Recently, morphological structure of digestive system and hematology of *Varanus salvator* have been investigated [31,32], however, the innate immunity against bacterial infection is rarely reported and still be controversial. Therefore, the purpose of this study is to examine the inhibition effect of pathogenic bacteria *in vitro* with crude peptides and protein hydrolysates extracted from *Varanus salvator*'s serum. This investigation would emerge the novel antimicrobial molecules for future medical development.

## Materials and methods

### Animals and sampling

The Asian water monitors, *Varanus salvator*, which more than 5 kilogram of body weight and more than 100 centimeters in a length of snout to anus were randomly assigned. The sex was not restricted. All Varanus had a good body condition status and no clinical sign (n = 21). The sampling procedure was done at Kao-Shon wildlife center in Ratchaburi province of Thailand. The animals used was approved by the Faculty of Veterinary Science-Animal Care and Use Committee (FVS-ACUC), Mahidol University, Thailand. (Protocol No. MUVS-2020-11-52). The methods were carried out in accordance with the approved guidelines.

Peripheral blood samples (10 ml.) were drawn from caudal tail vein of each *Varanus salvator*, using an 18-gauge needle into a plain blood collection tube. The whole blood was allowed to clot by leaving it undisturbed at room temperature for 30 minutes. All samples were maintained at 2–8°C while handling to the laboratory. All blood collections were removed the clot by centrifuging at 2,500 RPM for 10 minutes in a refrigerated centrifuge (4°C). Serum were collected and placed under -20°C for further analysis.

### Preparation of crude peptides and protein hydrolysate

The Varanus serum were diluted by five times with 0.5 M sodium acetate (NaOAc), then filtrated (cut-off) through a semipermeable membrane (Vivaspin 20, 3 kDa MWCO, GE Healthcare, Chicago, UK) to yield peptides smaller than 3 kDa, which were frozen at −20°C until use. The serum containing peptides and proteins with larger molecular size than 3 kDa were collected and protein concentration of these fractions was determined by Lowry's method using bovine serum albumin (BSA) as a standard protein [33]. Then the >3kDa peptide/protein were hydrolyzed by porcine pepsin (Sigma–Aldrich, St. Luis, MO, USA) at an enzyme/protein

ratio of 1:20 (mg/mg protein) at 37˚C for 16 hr. After terminating the enzymatic reaction by boiling for 10 min, the protein hydrolysate was filtered through a 3 kDa cut-off membrane (Vivaspin 20, 3 kDa MWCO, GE Healthcare, Chicago, UK). The protein hydrolysate samples were initially purified by reverse-phase chromatography using a Delta-Pak C18 column (100 Å, 3.9 mm × 150 mm; Interlink Scientific Services Ltd., Kent, UK) pre-equilibrated with 0.1% trifluoroacetic acid (TFA) in acetonitrile (ACN). The column was washed with 0.1% TFA in sterile water, after which a sample containing 0.1% TFA was loaded to bind the column. After washing with 0.1% TFA in sterile water, the hydrophobic fraction was eluted with 0.1% TFA in ACN. All steps were carried out at an adjusted flow rate of 1 mL/min. Both <3 kDa native peptides and <3 kDa protein hydrolysates bound to C18 column fractions (or hydrophobic protein hydrolysate) were further evaluated for antimicrobial activity.

## Antibacterial activities of crude peptides and protein hydrolysate

The selected bacteria including *Escherichia coli* ATCC25922, *Staphylococcus aureus* ATCC25923, Methicillin-resistant *Staphylococcus aureus* ATCC43300, *Enterobacter aerogenes* ATCC13048, *Bacillus subtilis* ATCC6633, *Bacillus cereus* ATCC11778, *Acinetobacter baumannii* ATCC19606, *Burkholderia cepacia* ATCC25416, *Pseudomonas aeruginosa* ATCC27853 and *Vibrio cholerae* ATCC51394 were used in this study. The 10% glycerol stock of bacteria (V/V) were kept at -80˚C and were reactivated in LB broth at 37˚C overnight before used. Antimicrobial activities of the protein hydrolysates and peptides against bacteria were determined using microdilution assay as previously described [34]. Briefly, the bacteria was grown to a final OD $_{600nm}$ of 0.8. Cultures will be then diluted in fresh broth to a cell density of ~ $1 \times 10^6$ colony forming units per milliliter (CFU/ml) and distributed into the wells of a 96-well microtiter plate (50 µl per well). The 50 µl protein hydrolysates and native peptides were prepared in fresh broth and added to cells (100 µl final volume) to make final screening concentrations of 100 µg/ml. The plates were incubated at 37˚C after which they will be measured at 600 nm for optical density (OD) after incubation for 6 h. Colistin was used as positive control while hydrolysate buffer was used as negative control. The inhibitory percentage was calculated from [(OD control − OD peptide test)/OD control] × 100 where OD control is the absorbance of control and OD peptide test is the absorbance of the sample.

## LC-MS analysis

The protein hydrolysate or crude peptide samples were prepared for injection into an Ultimate 3000 Nano/Capillary LC System (Thermo Scientific, UK) coupled to a Hybrid quadrupole Q-Tof impact II™ (Bruker Daltonics) equipped with a Nano-captive spray ion source. Briefly, 100 ng of protein hydrolysates or peptides were enriched on a µ-Precolumn 300 µm i.d. X 5 mm C18 Pepmap 100, 5 µm, 100 A (Thermo Scientific, UK), separated on a 75 µm I.D. x 15 cm and packed with Acclaim PepMap RSLC C18, 2 µm, 100Å, nanoViper (Thermo Scientific, UK). The C18 column was enclosed in a thermostatted column oven set to 60˚C. Solvent A and B containing 0.1% formic acid in water and 0.1% formic acid in 80% acetonitrile, respectively were supplied on the analytical column. A gradient of 5–55% solvent B was used to elute the peptides at a constant flow rate of 0.30 µl/min for 30 min. Electrospray ionization was carried out at 1.6kV using the CaptiveSpray. Nitrogen was used as a drying gas (flow rate about 50 l/h). Collision-induced-dissociation (CID) product ion mass spectra were obtained using nitrogen gas as the collision gas. Mass spectra (MS) and MS/MS spectra were obtained in the positive-ion mode at *2 Hz* over the range (m/z) 150–2200. The collision energy was adjusted to 10 eV as a function of the *m/z* value. The LC-MS analysis of each sample was done in triplicate.

MaxQuant 2.1.0.0 was used to quantify the proteins in individual samples using the Andromeda search engine to correlate MS/MS spectra to the Uniprot *Varanus salvator* database [35]. Label-free quantitation with MaxQuant's standard settings was performed: maximum of two miss cleavages, mass tolerance of 0.6 dalton for main search, unspecific digesting enzyme for peptide sample while pepsin for protein hydrolysate sample, and the oxidation of methionine as variable modifications. Only peptides with a minimum of 7 amino acids, as well as at least one unique peptide, were required for protein identification. Only proteins with at least two peptides, and at least one unique peptide, were considered as being identified and used for further data analysis. Protein FDR was set at 10% and estimated by using the reversed search sequences. The maximal number of modifications per peptide was set to 5. As a search FASTA file, the proteins present in the *Varanus salvator* proteome downloaded from Uniprot. Potential contaminants present in the contaminants.fasta file that comes with MaxQuant were automatically added to the search space by the software [35].

## Antimicrobial peptides (AMP) prediction

432 *Varanus salvator* peptides were submitted as fasta format to Antimicrobial Peptide Scanner vr.2 (https://www.dveltri.com/ascan/v2/ascan.html). Briefly, the server uses a deep neural network (DNN) as a prediction tool [36]. Specifically, the algorithm considers a peptide with a prediction probability > 0.5 as an AMP. In addition, isoelectric point (pI) and molecular weight were calculated using https://web.expasy.org/cgi-bin/compute_pi/pi_tool. All peptides were then calculated for hydrophobicity: hydrophilicity using Peptide2.com (Peptide Hydrophobicity/Hydrophilicity Analysis Tool. Selected peptides were modeled using PEP-FOLD Peptide Structure Prediction Server (univ-paris-diderot.fr) followed by PyMOL | pymol.org.

## Statistical analysis

Prior to conducting statistical analysis, the data was checked for normal distribution and homogeneity of variance. If any, a nonparametric test, the Wilcoxon rank-sum test, was performed to evaluate the median values of freely distributed continuous variables for inhibition. In addition, a one-sample T-test was applied to compare the mean values of inhibition. The analyses were conducted using PASW Statistics for Windows, version 18.0 [37]. Statistical significance was determined at a p-value of less than 0.05.

## Results

The crude peptides and protein hydrolysates derived from Varanus's serum showed some results on the selected bacteria. The crude peptides showed inhibition median values on *Staphylococcus aureus* ATCC25923 = 7.6%, Methicillin-resistant *Staphylococcus aureus* ATCC43300 = 5.7%, *Enterobacter aerogenes* ATCC13048 = 25.6%, *Bacillus cereus* ATCC11778 = 16.4%, *Acinetobacter baumannii* ATCC19606 = 33.4%, *Burkholderia cepacia* ATCC25416 = 35.3% and *Pseudomonas aeruginosa* ATCC27853 = 25.8% (Table 1). The inhibitory data of crude peptides derived from serum of all *Varanus salvator* (n = 21) was shown in S1 Table.

The protein hydrolysates showed inhibition median values on *Escherichia coli* ATCC25922 = 11.7%, *Staphylococcus aureus* ATCC25923 = 2.2%, *Enterobacter aerogenes* ATCC13048 = 6.4%, *Acinetobacter baumannii* ATCC19606 = 13.5%, *Burkholderia cepacia* ATCC25416 = 24.3% and *Pseudomonas aeruginosa* ATCC27853 = 12.0% (Table 1). The inhibitory data of protein hydrolysates derived from serum of all *Varanus salvator* (n = 21) was shown in S2 Table.

**Table 1. Inhibitory effect on 10 types of bacteria of crude peptides and protein hydrolysate derived from serum of *Varanus salvator* (n = 21).**

| Items | crude peptides | protein hydrolysate | p-value |
|---|---|---|---|
| | % inhibition | | |
| *Escherichia coli* ATCC25922 | 0.00[b] | 11.70[a] | 0.019 |
| *Staph. aureus* ATCC25923 | 7.60 | 2.20 | 0.472 |
| MR *Staph. Aureus* ATCC43300 | 5.70 [a] | 0.00 [b] | 0.002 |
| *Enterobacter aerogenes* ATCC13048 | 25.60 [a] | 6.40 [b] | 0.000 |
| *Bacillus subtilis* ATCC6633 | 0.00 | 0.00 | 0.180 |
| *Bacillus cereus* ATCC11778 | 16.40 [a] | 0.00 [b] | 0.000 |
| *Acinetobacter baumannii* ATCC19606 | 33.40 [a] | 13.50 [b] | 0.000 |
| *Burkholderia cepacia* ATCC25416 * | 35.30 [a] | 24.30 [b] | 0.000 |
| *Pseudomonas aeruginosa* ATCC27853 | 25.80 [a] | 12.00 [b] | 0.001 |
| *Vibrio cholerae* ATCC51394 | 0.00 | 0.00 | 0.109 |

[a,b] Median values with different superscript letters in the same row were statistically significant (Wilcoxon test, p-value < 0.05)

\* The data meets the criteria for parametric tests (one-sample T-test, p-value < 0.05).

Most crude peptides showed significant higher inhibitory effect than protein hydrolysates on most tested bacteria (p-value < 0.002) except *Escherichia coli* ATCC25922 which protein hydrolysates showed a significant higher effect (p-value = 0.019). Since the data was checked for normal distribution and homogeneity of variance, most data were decided to use non-parametric test, except *Burkholderia cepacia* ATCC25416. (Table 1).

Antimicrobial peptides prediction shows 41/432 sequences (above 0.5 probability). Table 2 shows 41 sequences above 0.5 probability. The hydrophobicity: hydrophilicity analysis demonstrates that they are highly hydrophobic especially AHPMPIPAWILMAM and AVWAFVV-CIPFFF. Most of them compose of hydrophobic, neutral, and basic amino acids, respectively (S3 Table). Therefore, they show quite high isoelectric point (pI) except DFWSQICSSW, DLYPTDPCCGYTV, EYVGWWTPSWVSQGY, CAPSEPFPVQ, and DNRPFYVECPS.

## Discussion

The antibacterial abilities of derived agent from *Varanus salvator* serum in the experiment indicated that the crude peptides appear to have more potential to be further investigated for antibacterial properties than protein hydrolysates.

To determine the antibacterial activity, most of crude peptides from *Varanus salvator* serum appear to be able to reduce the growth of Gram-negative representatives, including *Burkholderia cepacia* ATCC25416 (35.3%), *Acinetobacter baumannii* ATCC19606 (33.4%), *Pseudomonas aeruginosa* ATCC27853 (25.8%) and *Enterobacter aerogenes* ATCC13048 (25.6%). The rest, the Varanus's serum peptide seem unable to reduce the growth of *Escherichia coli* ATCC25922, *Vibrio cholerae* ATCC51394 and all Gram-positive representatives (Tables 1 and S1). According to a previous study in Komodo dragons, *Varanus komodoensis*, their serum show ability to inhibit growth of 5 Gram-negative (*Escherichia coli* ATCC25922, *Shigella flexnerii* ATCC700930, *Klebsiella oxytoca* ATCC49131, *Salmonella typhimurium* ATCC14028 and *Providencia stuartii* 33672) and 2 Gram-positive bacteria (*Staphylococcus aureus* ATCC6538 and *Streptococcus epidermitis* ATCC19615), which *Escherichia coli* ATCC25922 and *Staphylococcus aureus* ATCC6538 were moderately affected [19]. However, it was not the same Varanus species and was tested from pooled serum. Gram-negative bacteria seem to be affected by *Varanus salvator* serum peptides. To support this information, the antimicrobial peptide prediction also interpreted that the best hit sequence is highly hydrophobic, which have ability to destroy Gram-negative outer membrane, lead to reducing the bacterial properties.

**Table 2. Antimicrobial peptide prediction of *Varanus salvator* serum, using Antimicrobial Peptide Scanner vr.2 (https://www.dveltri.com/ascan/v2/ascan.html).**

| Sequence | Length | MW (kDa) | pI | Hydrophobicity (%) | Probability |
|---|---|---|---|---|---|
| AAIMNWKLCAQLAAFCWGSSFM | 22 | 2.45 | 8.10 | 63.64 | 0.9989 |
| INHFFCDTPALLKATCS | 17 | 1.88 | 6.73 | 47.06 | 0.9913 |
| CILPLCGWGTYASTS | 15 | 1.57 | 5.51 | 40 | 0.9862 |
| AMLHTCGTFANTFCS | 15 | 1.60 | 6.76 | 40 | 0.9833 |
| CKYKGPSTQGCVLN | 14 | 1.50 | 8.86 | 21.43 | 0.9793 |
| AVWAFVVCIPFFF | 13 | 1.55 | 5.56 | 92.31 | 0.9725 |
| CSFPFIYKGKTYTECTS | 17 | 1.98 | 8.02 | 23.53 | 0.9649 |
| CCLNPILYAF | 10 | 1.16 | 5.51 | 60 | 0.9625 |
| MGLLSTMVGGFGGLN | 15 | 1.45 | 5.28 | 46.67 | 0.9589 |
| AHPMPIPAWILMAM | 14 | 1.58 | 6.79 | 92.86 | 0.9578 |
| GSTDKSPWCATTSNYDRDRKWKPCA | 25 | 2.87 | 8.82 | 24 | 0.9463 |
| CPVDQTGYRDMRCRN | 15 | 1.81 | 8.06 | 20 | 0.9447 |
| AFLWFGCLMAF | 11 | 1.31 | 5.56 | 81.82 | 0.9307 |
| ACGGWLRRHAI | 11 | 1.24 | 10.35 | 45.45 | 0.9285 |
| CLPNSACVQTSPG | 13 | 1.28 | 5.51 | 38.46 | 0.9154 |
| GSGCGLGSTSGIRDLRNGFCGSGP | 24 | 2.25 | 8.07 | 20.83 | 0.9011 |
| CAINLCPNEPLKYFLVCQYCPG | 22 | 2.49 | 5.97 | 45.45 | 0.9001 |
| DFWSQICSSW | 10 | 1.26 | 3.80 | 40 | 0.8892 |
| FDPLGSARLPFSLHFF | 16 | 1.85 | 6.74 | 62.5 | 0.8569 |
| EYVGWWTPSWVSQGY | 15 | 1.84 | 4.00 | 40 | 0.8472 |
| CPIQCNAQQTGPWTSAKS | 18 | 1.92 | 8.06 | 33.33 | 0.8452 |
| AKMGFPFRRG | 10 | 1.17 | 12.01 | 50 | 0.8221 |
| GLWAMVWHHST | 11 | 1.32 | 6.92 | 54.55 | 0.8167 |
| ACFGFGMVAGP | 11 | 1.06 | 5.56 | 63.64 | 0.7887 |
| QTQNLGGFGGVMTS | 14 | 1.40 | 5.52 | 28.57 | 0.7861 |
| GGYPCGQPMMPGVYT | 15 | 1.56 | 5.52 | 40 | 0.7844 |
| GGSGNPSHKPRS | 12 | 1.18 | 11.00 | 16.67 | 0.784 |
| CLFFLSAGNAHLNRLLW | 17 | 1.97 | 8.26 | 58.82 | 0.7045 |
| DLYPTDPCCGYTV | 13 | 1.45 | 3.56 | 30.77 | 0.6956 |
| APQEYTHYPPPCG | 13 | 1.46 | 5.24 | 38.46 | 0.6782 |
| CQNASVFNTGAAAAAAAH | 18 | 1.67 | 6.73 | 55.56 | 0.6556 |
| CAPSEPFPVQ | 10 | 1.07 | 4.00 | 60 | 0.6318 |
| CPDARVMLNTTCTSGKS | 17 | 1.78 | 8.06 | 29.41 | 0.628 |
| CPPFKPDVYNSNI | 13 | 1.49 | 5.83 | 46.15 | 0.6259 |
| AMLCGFWLFAQS | 12 | 1.37 | 5.56 | 66.67 | 0.6076 |
| AYPWWHMTDYQLCAGILGGGRDTC | 24 | 2.71 | 5.21 | 37.5 | 0.5945 |
| ERCYQGTSNR | 10 | 1.21 | 8.32 | 0 | 0.5943 |
| ACFSWKKDKDYNATTACWFI | 20 | 2.40 | 8.06 | 40 | 0.5925 |
| DNRPFYVECPS | 11 | 1.33 | 4.37 | 36.36 | 0.5764 |
| AFMAYFYLQTFCHAF | 15 | 1.86 | 6.77 | 60 | 0.5469 |
| ACMRDSGGPLLC | 12 | 1.22 | 5.86 | 41.67 | 0.5129 |

The low percentages of peptide's inhibition effect can be explained since it was the screening test from crude peptides which may compose of both active and inactive peptides. Therefore, the experimental results show the possibility that Varanus serum should composes of certain active peptides. The scanning results of peptide sequences also support these findings by 41 of 432 sequences were interpreted as the potential antimicrobial peptides derived from

Asian water monitor (*Varanus salvator*) serum. From the previous study, the 48 potential antimicrobial peptides had been identified from Komodo dragon (*Varanus komodoensis*) plasma, which 8 of these peptides against *Pseudomonas aeruginosa* ATCC9027 and *Staphylococcus aureus* ATCC25923 had been evaluated [20]. Therefore, these 41 sequences from *Varanus salvator* would make further study confidence.

Antimicrobial peptides (AMPs) have recognized as the multi-functional peptides which have an important role in the immediate defenses against variety of pathogenic microorganisms, including Gram-negative and Gram-positive bacteria, fungi, viruses and parasites [2,5,9,38]. According to our results, 41 potential AMPs identified along with the antibacterial inhibition evidence indicate the possibility to discover the effective antibacterial peptides from *Varanus salvator* in the next step. In particular, high isoelectric point (PI > 8) with highly hydrophobicity of AMPs sequences supports the experiments that all bacteria being affected are Gram-negative. Moreover, those amino acids characterized as neutral and may require organic solvents for better solution (S3 Table). Typically, the AMP-mediated microbial killing usually occurs through membrane permeability. Based on ionic and hydrophobic interactions, it provides AMPs interact with Gram-negative cell envelope which compose of phospholipids in the inner leaflet and lipopolysaccharides (LPS) in the outer leaflet [39,40]. The peptides penetrate both cell envelope membranes and then kill cells by a multi-mechanism that involves action on more than one anionic target [5,9]. Interestingly, high ordered of secondary structure of four peptides; AAIMNWKLCAQLAAFCWGSSFM, INHFFCDTPALLKATCS, CILPLCGWGTYASTS and AMLHTCGTFANTFCS may stabilize these peptides in serum (S1 Fig). Moreover, most of the studies on AMPs reveal that, instead of antibiotic, peptides treated can prevent endotoxin (LPS) released, and not allow opportunity for bacteria to develop resistance [2,6].

According to, two Gram-negative representatives, *Escherichia coli* ATCC25922 and *Vibrio cholerae* ATCC51394, could not be affected by Varanus serum crude peptides, the study of modification of peptide synthesis may be required optimization in the future. The peptide hybridization has been studied to enhance antibacterial peptide performance to the membrane permeabilization [41,42]. For example, link a C12-alkyl chain or a Lp-I to the C-terminus of Bac7(1–16), Bac-C12 and Bac-Lp-I respectively, could enhance the activity against more bacterial species. Moreover, unlike Bac7(1–16), Bac-C12 and Bac-Lp-I did not select resistant mutants in *E. coli* [41]. Thus, the antibacterial peptides derived from living organisms, especially the *Varanus salvator*, should be more investigated.

## Conclusions

In conclusion, crude peptides derived from Asian water monitor (*Varanus salvator*) serum have antibacterial potential against *Enterobacter aerogenes* ATCC13048, *Acinetobacter baumannii* ATCC19606, *Burkholderia cepacia* ATCC25416 and *Pseudomonas aeruginosa* ATCC27853. The 41 potential antimicrobial peptides have been identified. From the prediction, most AMPs are highly hydrophobic, which is in accordance with the *in vitro* inhibition of Gram-negative bacteria.

The results provide opportunity for novel antibacterial agent to be discovered. Furthermore, searching for antimicrobial peptide candidate should be the next step. The potential of AMPs obtained from *Varanus salvator* serum will be the starting point for peptide synthesis and further modification to increase the antibacterial efficiency in the future.

## Supporting information

**S1 Fig. 3D structures of Antimicrobial peptides from Varanus salvator serum were predicted using PEP-FOLD.** Model were generated by PyMOL. A)

AAIMNWKLCAQLAAFCWGSSFM, B) INHFFCDTPALLKATCS, C) CILPLCGWG-
TYASTS, D) AMLHTCGTFANTFCS.
(PDF)

**S1 Table. Inhibitory effect on 10 types of bacteria of crude peptides derived from serum of
Varanus salvator (n = 21).**
(PDF)

**S2 Table. Inhibitory effect on 10 types of bacteria of protein hydrolysates derived from
serum of Varanus salvator (n = 21).**
(PDF)

**S3 Table. Amino acid properties of 41 Antimicrobial peptides derived from Varanus salva-
tor serum.**
(PDF)

## Acknowledgments

The authors gratefully acknowledge Mr.Pakpoom Aramsirirujiwet for forestry technical exper-
tise at Khao-son Wildlife Breeding Station, the Department of National Park Wildlife and
Plant Conservation for the sampling assistance and Asst. Prof. Dr. Surasak Jittakhot for the sta-
tistical consultation.

## Author Contributions

**Investigation:** Jitkamol Thanasak, Sittiruk Roytrakul, Waraphan Toniti, Janthima
Jaresitthikunchai, Narumon Phaonakrop, Siriwan Thaisakun, Sawanya Charoenlappanit,
Rudee Surarit, Wanna Sirimanapong.

**Project administration:** Jitkamol Thanasak.

**Writing – original draft:** Jitkamol Thanasak.

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
