## [Decision Letter · Decision Letter 0]

14 Jun 2023

PONE-D-23-13258The Investigation of Antibacterial Properties of Peptides and Protein Hydrolysates Derived from Serum of Asian Water Monitor (Varanus salvator)PLOS ONE

Dear Dr. Thanasak,

Thank you for submitting your manuscript to PLOS ONE. After careful consideration, we feel that it has merit but does not fully meet PLOS ONE’s publication criteria as it currently stands. Therefore, we invite you to submit a revised version of the manuscript that addresses the points raised during the review process.

We look forward to receiving your revised manuscript.

Kind regards,

Md. Imtaiyaz Hassan, Ph.D.

Academic Editor

PLOS ONE

Journal Requirements:

https://www.mdpi.com/1422-0067/22/15/7959

https://www.frontiersin.org/articles/10.3389/fmicb.2021.771527/full

In your revision ensure you cite all your sources (including your own works), and quote or rephrase any duplicated text outside the methods section. Further consideration is dependent on these concerns being addressed.

Additional Editor Comments:

A critical assessment of the present knowledge with some clear conclusions what all these results mean, and directions for future research and potential applications should be strengthened.

I found minor issue, many typos and grammatical errors are seen in the paper. There are grammatical mistakes and typographical errors in the manuscript. The author should recheck this manuscript carefully and remove all such errors.

Reviewers' comments:

Reviewer's Responses to Questions

**Comments to the Author**

1. Is the manuscript technically sound, and do the data support the conclusions?

Reviewer #1: Yes

Reviewer #2: Yes

2. Has the statistical analysis been performed appropriately and rigorously? 

Reviewer #1: Yes

Reviewer #2: Yes

3. Have the authors made all data underlying the findings in their manuscript fully available?

Reviewer #1: Yes

Reviewer #2: Yes

4. Is the manuscript presented in an intelligible fashion and written in standard English?

Reviewer #1: Yes

Reviewer #2: Yes

5. Review Comments to the Author

Reviewer #1: The authors demonstrated that "antimicrobial peptides were predicted which shows highly hydrophobic". It will be interesting to test and check their amyloidogenicity according the last review for AMPs - Int J Mol Sci. 2022 May 13;23(10):5463. doi: 10.3390/ijms23105463.

Line 45. “ceropin” correct to “cecropin”.

Lines 59-60. “Moreover, BmKn-2, an animal peptide especially extracted from scorpion venom, has shown the inhibitory effect on bacterial growth [11, 14-16].” I think that BmKn-2 is not discussed in the cited work #15. Please check.

Line 101. “digestive System” correct to “digestive system”.

Line 117. “a plain blood collecting tube” correct to “a plain blood collection tube”.

Lines 156-157. “The inhibitory percentage was calculated from [(OD control − OD peptide test)/OD control] × 100”. OD control – is it OD for positive or negative control? If this is the optical density for the negative control, then Table 1 should indicate the antimicrobial efficacy of the crude peptides and protein hydrolysates in comparison with the antibiotic, colistin.

Reviewer #2: The manuscript titled as “The Investigation of Antibacterial Properties of Peptides and Protein Hydrolysates Derived from Serum of Asian Water Monitor (Varanus salvator)” represents an interesting study and is well written that only need minor modifications. However, I feel that some relevant images can be added to make the study more attractive and interesting. Please consider the following points:

1. Title and abstract:

Title and abstract are accurate and adequately addressed. However, a graphical abstract can be added.

2. Introduction:

1. L- 42: Please give the examples of resistant and non-resistant pathogenic bacteria.

2. L-73: ‘evidence of reduced in bacterial growth’ these sentence needs to be re-written.

3. L-80: ‘according to the list of Thailand protected species’, please give an appropriate reference for this sentence.

3. Materials and Methods

1. L-27: Why Lowry's method is selected for protein determination over other methods?

2. L-130: Why boiling for 10 minutes is done?

3. L-155: ‘Colistin was used as positive control while hydrolysate buffer was used as negative control’. Please specify reason for this statement.

4. L-156: mention complete formula in mathematical manner.

5. L-169: Specify the reason for the use of nitrogen as a drying gas.

4. Results:

1. Table 2. Antimicrobial peptide prediction of Varanus salvator serum, using’ this line seems incomplete.

2. L-247 ‘5 Gram-negative and 2 Gram-positive bacteria,’ please mention the names of concerned bacteria and also mention reference for this statement.

3. L-269: ‘high isoelectric point’ please mention the numerical value also.

4. L-273 ‘some mechanisms can be proposed’ give brief description of the mechanisms that can be proposed.

5. Conclusion:

1. L-294 Antimicrobial peptides…..Hydrophobic, this line is not in flow with this paragraph, so needs to be re-written.

2. Conclusion section is written in brief. Please add more relevant points in it.

3. Also write a concluding line in the end with future scope of this research.

6. PLOS authors have the option to publish the peer review history of their article (what does this mean?). If published, this will include your full peer review and any attached files.

Reviewer #1: No

Reviewer #2: No

---

## [Author Response · Author response to Decision Letter 0]

11 Jul 2023

Dear reviewers

According to some points raised by the academic reviewers to the manuscript entitled “The Investigation of Antibacterial Properties of Peptides and Protein Hydrolysates Derived from Serum of Asian Water Monitor (Varanus salvator)”. Please see the responses (the answers in blue) to each point which be sent to you in a Response to Reviewers file. Many thanks for all the helpful points.

Your sincerely,

The authors

---

## [Decision Letter · Decision Letter 1]

22 Aug 2023

PONE-D-23-13258R1The Investigation of Antibacterial Properties of Peptides and Protein Hydrolysates Derived from Serum of Asian Water Monitor (Varanus salvator)PLOS ONE

Dear Dr. Thanasak,

Thank you for submitting your manuscript to PLOS ONE. After careful consideration, we feel that it has merit but does not fully meet PLOS ONE’s publication criteria as it currently stands. Therefore, we invite you to submit a revised version of the manuscript that addresses the points raised during the review process.

We look forward to receiving your revised manuscript.

Kind regards,

Md. Imtaiyaz Hassan, Ph.D.

Academic Editor

PLOS ONE

Journal Requirements:

Additional Editor Comments:

Authors must address all the comments.

Reviewers' comments:

Reviewer's Responses to Questions

**Comments to the Author**

1. If the authors have adequately addressed your comments raised in a previous round of review and you feel that this manuscript is now acceptable for publication, you may indicate that here to bypass the “Comments to the Author” section, enter your conflict of interest statement in the “Confidential to Editor” section, and submit your "Accept" recommendation.

Reviewer #1: (No Response)

2. Is the manuscript technically sound, and do the data support the conclusions?

Reviewer #1: Yes

3. Has the statistical analysis been performed appropriately and rigorously? 

Reviewer #1: (No Response)

4. Have the authors made all data underlying the findings in their manuscript fully available?

Reviewer #1: Yes

5. Is the manuscript presented in an intelligible fashion and written in standard English?

Reviewer #1: (No Response)

6. Review Comments to the Author

Reviewer #1: The authors did not answer on my previous question: "The authors demonstrated that "antimicrobial peptides were predicted which shows highly

hydrophobic". It will be interesting to test and check their amyloidogenicity according the last

review for AMPs - Int J Mol Sci. 2022 May 13;23(10):5463. doi: 10.3390/ijms23105463."

7. PLOS authors have the option to publish the peer review history of their article (what does this mean?). If published, this will include your full peer review and any attached files.

Reviewer #1: No

---

## [Author Response · Author response to Decision Letter 1]

20 Sep 2023

Reviewer #1: 

The authors demonstrated that "antimicrobial peptides were predicted which shows highly hydrophobic". It will be interesting to test and check their amyloidogenicity according the last review for AMPs - Int J Mol Sci. 2022 May 13;23(10):5463. doi: 10.3390/ijms23105463.

Response from authors

Based on the reviews you mentioned [1], we had run for amyloidogenic antimicrobial peptide (AAMP) prediction on the AMPs presented in Table 2, using iAMY-SCM : http://camt. pythonanywhere.com/iAMY-SCM at threshold = 288.5625 [2, 3]. We found that 14 sequences of 41 AMPs were showed AAMP properties, however, only 5 AAMPs were characterized to be the peptides originated from Varanus spp., including NADH dehydrogenase subunit 2 (mitochondrion); olfactory receptor, partial; olfactory receptor 1052-like; kallikrein-Vgou1 and ESP-Vind2, partial. The rest are uncharacterized proteins or belonged to other species such as bacteria and virus. 

Since we used peptides with low MW (3 kDa or smaller), therefore, the results of polymerized AAMP could not be tested in this experiment. However, the mechanism of action of predicted AAMP is interesting for further in-depth study.

References

1. Galzitskaya OV, Kurpe SR, Panfilov AV, Glyakina AV, Grishin SY, Kochetov AP, et al. Amyloidogenic Peptides: New Class of Antimicrobial Peptides with the Novel Mechanism of Activity. International Journal of Molecular Sciences [Internet]. 2022; 23(10).

2. Teng Z, Zhang Z, Tian Z, Li Y, Wang G. ReRF-Pred: predicting amyloidogenic regions of proteins based on their pseudo amino acid composition and tripeptide composition. BMC Bioinformatics. 2021;22(1):545. doi: 10.1186/s12859-021-04446-4.

3. Charoenkwan P, Kanthawong S, Nantasenamat C, Hasan MM, Shoombuatong W. iAMY-SCM: Improved prediction and analysis of amyloid proteins using a scoring card method with propensity scores of dipeptides. Genomics. 2021;113(1, Part 2):689-98. doi: https://doi.org/10.1016/j.ygeno.2020.09.065.

---

## [Editor Report · Decision Letter 2]

3 Oct 2023

The Investigation of Antibacterial Properties of Peptides and Protein Hydrolysates Derived from Serum of Asian Water Monitor (Varanus salvator)

PONE-D-23-13258R2

Dear Dr. Thanasak,

We’re pleased to inform you that your manuscript has been judged scientifically suitable for publication and will be formally accepted for publication once it meets all outstanding technical requirements.

Kind regards,

Md. Imtaiyaz Hassan, Ph.D.

Academic Editor

PLOS ONE

Additional Editor Comments (optional):

The revised submission is acceptable now.
---

## [Editor Report · Acceptance letter]

10 Oct 2023

PONE-D-23-13258R2 

The Investigation of Antibacterial Properties of Peptides and Protein Hydrolysates Derived from Serum of Asian Water Monitor (Varanus salvator) 

Dear Dr. Thanasak:

I'm pleased to inform you that your manuscript has been deemed suitable for publication in PLOS ONE. Congratulations! Your manuscript is now with our production department. 

Kind regards, 

on behalf of

Dr. Md. Imtaiyaz Hassan 

Academic Editor

PLOS ONE